# Ribonuclease L mediates the cell-lethal phenotype of double-stranded RNA editing enzyme ADAR1 deficiency in a human cell line

Yize Li[1†], Shuvojit Banerjee[2†], Stephen A Goldstein[1], Beihua Dong[2], Christina Gaughan[2], Sneha Rath[3], Jesse Donovan[3], Alexei Korennykh[3], Robert H Silverman[2*], Susan R Weiss[1*]

[1]Department of Microbiology, Perelman School of Medicine, University of Pennsylvania, Philadelphia, United States; [2]Department of Cancer Biology, Lerner Research Institute, Cleveland Clinic, Cleveland, United States; [3]Department of Molecular Biology, Princeton University, Princeton, United States

**Abstract** ADAR1 isoforms are adenosine deaminases that edit and destabilize double-stranded RNA reducing its immunostimulatory activities. Mutation of *ADAR1* leads to a severe neurodevelopmental and inflammatory disease of children, Aicardi-Goutiéres syndrome. In mice, *Adar1* mutations are embryonic lethal but are rescued by mutation of the *Mda5* or *Mavs* genes, which function in IFN induction. However, the specific IFN regulated proteins responsible for the pathogenic effects of *ADAR1* mutation are unknown. We show that the cell-lethal phenotype of *ADAR1* deletion in human lung adenocarcinoma A549 cells is rescued by CRISPR/Cas9 mutagenesis of the *RNASEL* gene or by expression of the RNase L antagonist, murine coronavirus NS2 accessory protein. Our result demonstrate that ablation of RNase L activity promotes survival of ADAR1 deficient cells even in the presence of MDA5 and MAVS, suggesting that the RNase L system is the primary sensor pathway for endogenous dsRNA that leads to cell death.

*For correspondence: silverr@ccf. org (RHS); weisssr@upenn.edu (SRW)

†These authors contributed equally to this work

Competing interests: The authors declare that no competing interests exist.

## Introduction

Adenosine deaminase acting on dsRNA (ADAR1) isoforms neutralize the immunostimulatory activity of double-stranded RNA (dsRNA) by converting adenosine to inosine (i.e., A-to-I editing) (*George et al., 2014*; *Tomaselli et al., 2015*; *Patterson and Samuel, 1995*). ADAR1 alters the RNA coding sequence, because I is read as G, and it also destabilizes dsRNA because A:U base pairs are disrupted than I:U mismatches. These modifications by ADAR1 suppress the cellular responses to dsRNA and prevent to type I IFN production. There are three mammalian ADAR genes (ADAR1-3), but only ADAR1 isoforms p110 and p150 and ADAR2 have deaminase activity (*Tomaselli et al., 2015*). The IFN inducible ADAR1 p150 isoform (subsequently 'p150') is present in both the cytoplasm and nucleus, while the constitutive ADAR1 p110 (an N-terminal truncated isoform translated from a shorter mRNA initiated at a downstream promoter in *ADAR1*) and ADAR2 are mostly nuclear. Mutations of *ADAR1* result in the severe, sometimes lethal, childhood neurodevelopmental disease, Aicardi-Goutiéres syndrome (*Rice et al., 2012*).

Interestingly, ADAR1 can be either pro-viral or anti-viral depending on the virus-host cell context (reviewed in [*George et al., 2014*]). The antiviral effects are due to hyper-editing and mutagenesis of viral RNAs (*Samuel, 2011*). Proviral effects are due in part to editing of viral RNAs (*Wong and Lazinski, 2002*) and/or to destabilizing dsRNA resulting in suppression of dsRNA-signaling through

MDA5 and MAVS to type I IFN genes (*Figure 1*). Accordingly, mutation of either MDA5 or MAVS rescues the embryonic lethal phenotype of *Adar1* knockout (KO) mice (*Pestal et al., 2015*; *Liddicoat et al., 2015*; *Mannion et al., 2014*). ADAR1 also antagonizes the IFN-inducible dsRNA-dependent serine/threonine protein kinase, PKR, presumably by altering the structure of dsRNA and thereby preventing both PKR activation and phosphorylation of its substrate protein, eIF2α (*Samuel, 2011*; *Gélinas et al., 2011*; *Wang et al., 2004*). However, whereas effects of ADAR1 on PKR activity have been extensively studied, ADAR1 effects on another IFN-regulated dsRNA-activated antiviral pathway, the oligoadenylate-synthetase (OAS-RNase L) system, have not been described. OAS isoforms (OAS1, OAS2, OAS3) are IFN inducible enzymes that sense dsRNA and produce 2′,5′-oligoadenylates (2-5A) which activate RNase L to degrade viral and host single-stranded RNAs leading to apoptosis and inhibition of virus growth (*Silverman and Weiss, 2014*). Here we report that whereas *ADAR1* single gene KO A549 cells were not viable, it was possible to rescue *ADAR1* deficient cells by knockout (KO) of either *RNASEL* or *MAVS* or by expression of a viral antagonist of the OAS/RNase L system (*Silverman and Weiss, 2014*). Our results suggest that the RNase L activation is the primary mode of cell death induced by either endogenous or exogenous dsRNA.

## Results

### RNase L activity is the major pathway leading to dsRNA-induced cell death

Before assessing the role of ADAR in regulating the RNase L pathway we compared the roles of MAVS, RNase L and PKR in mediating dsRNA induced cell death in A549 cells. Thus we used lentivirus delivered CRISPR/Cas9 and single-guide (sg)RNA (*Table 1*) to construct A549 cell lines with disruption of genes encoding each of these proteins, *RNASEL* KO, *MAVS* KO, *PKR* KO cells as well as *MAVS-RNASEL* double knockout (DKO). Disruption of each gene and protein expression in the absence or presence of IFN-α was confirmed by sequence analysis and Western immunoblot (*Figure 2a–c*; *Table 2*). The various A549 mutant cell lines were characterized for their sensitivity or resistance to exogenous dsRNA by poly(rI):poly(rC) (pIC) transfection as compared to wild type (WT) A549 (*Figure 3*). We initially transfected WT A549 and KO with a range of concentrations of pIC and

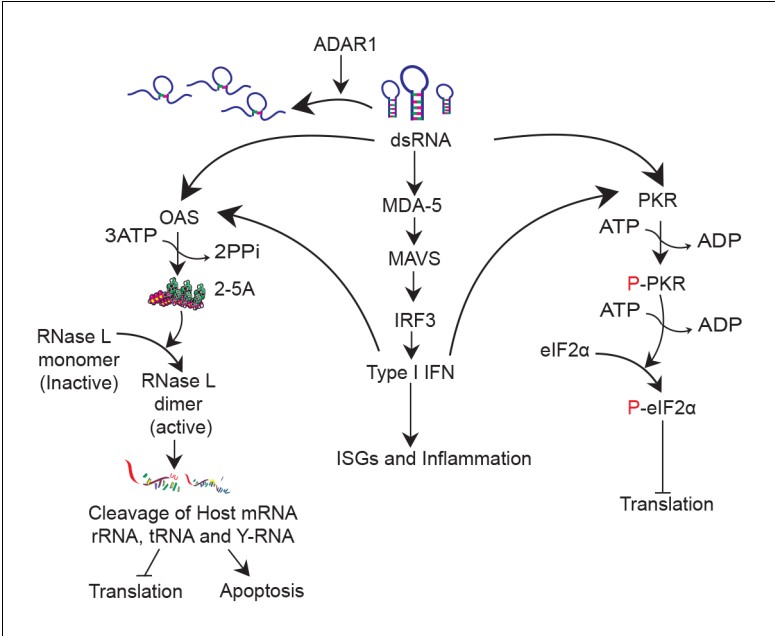

**Figure 1.** DsRNA induced antiviral pathways. DsRNA can be destabilized by ADAR1 activity. In the absence of ADAR1 dsRNA can be recognized by (1) MDA5 leading to IFN production; (2) OAS leading to activation of RNase L and eventually translational inhibition and apoptosis and (3) PKR leading to inhibition of translation.

**Table 1.** Construction of the plasmids for knockout of human ADAR1, MAVS and PKR using CRISPR/Cas9.

| Genes | Primers | | Nucleotide sequences (5'−3')* | Targeting Region |
|-------|---------|---|-------------------------------|------------------|
| ADAR1 | sgADAR1-4 | Forward | CACCG*TCTGTCAAATGCCATATGGG* | Exon2 |
| | | Reverse | AAAC*CCCATATGGCATTTGACAGA*C | |
| | sgADAR1-6 | Forward | CACCG*ACTCAGTTCCTGGAAATGTG* | Exon2 |
| | | Reverse | AAAC*CACATTTCCAGGAACTGAGT*C | |
| MAVS | sgMAVS-1 | Forward | CACCG*GAGGGCTGCCAGGTCAGAGG* | Exon4 |
| | | Reverse | AAAC*CCTCTGACCTGGCAGCCCT*C | |
| PKR | sgPKR-1 | Forward | CACCG*TAATACATACCGTCAGAAGC* | Exon1 |
| | | Reverse | AAAC*GCTTCTGACGGTATGTATTA*C | |

*Nucleotides sequences which were used for targeting gene are indicated as bold and italic.

at 48 hr post treatment cells were fixed and stained with crystal violet. Cells lacking RNase L expression were resistant to cell death at treatment with up to 5 µg/ml of pIC while treatment of WT A549 as well as PKR KO or MAVS KO cells with 0.5 µg/ml of pIC promoted cell death (*Figure 3a*). To obtain a more quantitative measure of cell death as well as to assess the effects of ADAR1 ablation on cell death, we compared the kinetics of pIC-induced cell death with the same set of cells in real time with an IncuCyte Live Cell Imaging System and software (*Figure 3b*). We also observed markers of apoptosis (caspase 3/7 activity) in real time (*Figure 3c*). While cells expressing RNase L died by 24–30 hr post treatment, cells ablated for RNase L expression (*RNASEL* KO and *RNASEL-MAVS* DKO) were resistant to pIC induced cell death (*Figure 3b&c*). Similarly, *RNASEL* KO or *RNASEL-MAVS* DKO showed accelerated wound healing in monolayer scratch assays compared with *MAVS* KO and WT cells, all of which retained RNase L (*Figure 3d*). RNase L was previously reported to impair cell migration in these assays (*Banerjee et al., 2015*; *Rath et al., 2015*) as well as to promote apoptosis (*Zhou et al., 1997*). Finally, the protective effect of *RNASEL* KO was not restricted to A549 cells as human mammary epithelial (HME) cells were also protected from pIC-induced cell death by ablation of RNase L expression (*Figure 3e*).

## Rescue of *ADAR1* KO cells by KO of *RNASEL* or *MAVS*

Mouse studies had shown that the embryonic lethal phenotype of *Adar1* KO could be rescued by KO of *Mavs* or *Mda5* presumably by reducing the downstream IFN signaling effects of dsRNA produced in the absence of ADAR1 editing (*Pestal et al., 2015*). We found that *ADAR1* KO is also lethal in A549 cells (*Table 3* and described below). Since activation of RNase L by exogenously provided pIC accelerated cell death so dramatically in A549 cells, we investigated whether KO of *RNASEL*, as well as dsRNA sensors MAVS and PKR, could rescue A549 *ADAR1* KO cells from lethal the effects of dsRNA. Using lentivirus delivered CRISPR/Cas9 and sgRNA4 directed at Exon 1 of *ADAR1* within the coding regions of p110 and p150 (*Table 1*; *Figure 2—figure supplement 1*) we screened more than 40 clones derived from WT A549 cells or PKR KO A549 and were unable to recover cells with a KO of ADAR1 (*Table 3*). The WT or PKR KO clones in which *ADAR1* remained intact were apparently derived from cells in which the CRISRP/Cas9 targeting vector failed to mutate ADAR1. However using *RNASEL* KO or *MAVS* KO A549 cells and similar lentivirus delivery of sgRNA4 we were able to recover both *RNASEL-ADAR1* and *MAVS-ADAR1* double KO (DKO) cells (*Table 3*, *Figure 2*). We compared the kinetics of cell death in response to pIC transfection of these DKO cells using the IncuCyte system (*Figure 4a*). While MAVS-ADAR1 DKO cells died more slowly than WT cells, RNase L-ADAR1 DKO cells were almost completely protected from cell death further supporting the finding that RNase L is the major pathway leading to cell death in the absence of ADAR1 editing. As might be expected, the kinetics of cell death following treatment with exogenous pIC (not a substrate of ADAR1) are similar in *RNASEL* KO and *RNASEL-ADAR1* DKO or *MAVS* KO and *MAVS-ADAR1* DKO (compare *Figure 3b* and *Figure 4a*).

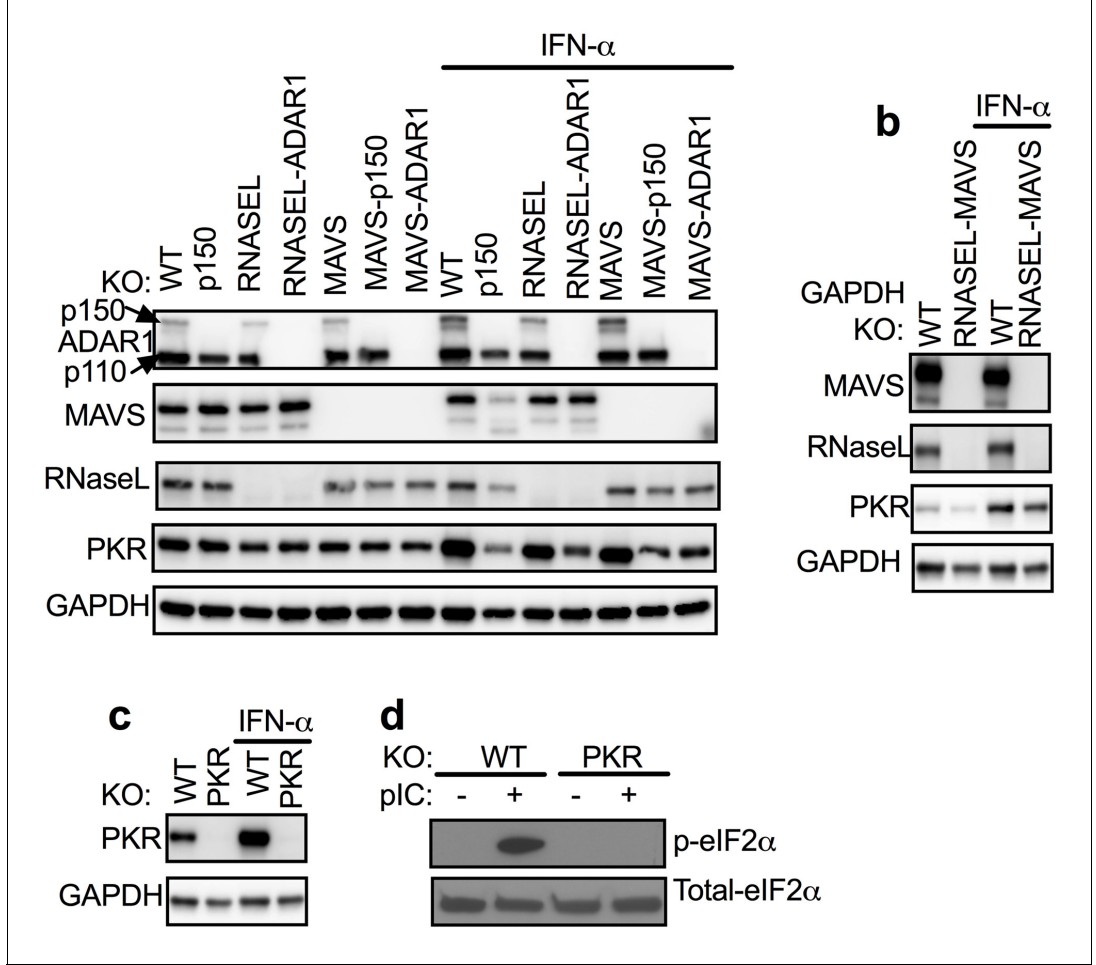

**Figure 2.** *ADAR1* KO cells were rescued from *RNASE L* KO or *MAVS* KO but not from WT or *PKR* KO A549 cells. (a) WT, WT-p150 KO, *RNASEL* KO, *RNASEL-ADAR* DKO, *MAVS* KO, *MAVS*-p150 DKO, *MAVS-ADAR1* DKO; (PKR expression was induced 1.6–1.9 fold following IFN treatment in cells expressing ADAR1 and reduced 0.3–0.6 fold in cells deleted for ADAR1 or ADAR1 p150; *Figure 2—source data 1*.) (b) WT or *RNASEL-MAVS* DKO or (c) WT or *PKR* KO cells were treated or mock treated with 1000 U/ml IFN-α overnight, lysed and proteins analyzed by Western immunoblotting with antibodies as indicated. (d) WT and *PKR* KO cells were transfected with pIC (1 µg/ml). After 2 hr, proteins in cell lysates were separated by 12% SDS/PAGE, transferred to PVDF-membranes, and probed with antibodies to detect total eIF2α or phosphorylated eIF2α. Immunoblots (in a–c) were performed at least two times and one representative blot is shown. See *Figure 2—figure supplement 1*.

The following source data and figure supplement are available for figure 2:

**Source data 1.** Quantification of PKR induction (+/−- IFN) treatment for *Figure 2a*.

**Figure supplement 1.** Schematic diagram of sgRNAs targeting the *ADAR1* gene.

## RNase L promotes lethality of *ADAR1* KO cells in the absence of exogenous dsRNA

In order to directly demonstrate that activation of RNase L leads to the cell-lethal phenotype of *ADAR1* KO cells, we transduced *RNASEL* KO and *RNASEL-ADAR1* DKO cells with lentiviruses expressing either WT RNase L or inactive mutant RNase L[R667A] (*Dong et al., 2001*). (Expression of RNase L and RNase L[R667A] by Western immunoblot is shown in *Figure 4b*.) Transduced cells were incubated and analyzed for total and dead cells over a period of time using the IncuCyte system (*Figure 4c*). WT RNase L expression in *RNASEL-ADAR1* DKO caused massive cell death while expression of inactive mutant RNase L did not. Also, expression of WT or mutant RNase L had no

**Table 2.** Primers for genotyping ADAR1, MAVS, and PKR knockout cells.

| Genes | Primers* | Nucleotide sequences (5′−3′) |
|---|---|---|
| ADAR1 | Forward-1 | ACCTTCCCTCCCAGGACTCCGGCC |
| | Reverse-1 | CCTGAGTGGAGACCGCGATTTTCC |
| | Forward-2 | ATGGCCGAGATCAAGGAGAAAATC |
| | Reverse-2 | GTTCTGGTCTGGCCTCTTGCCTG |
| MAVS | Forward | CTCCCCTGGCTCCTGTGCTCC |
| | Reverse | AACTCCCTTTATTCCCACCTTG |
| PKR | Forward | AGACTGAGATGAGTCCTATAAAG |
| | Reverse | TCACCTATGAGTGAGAACATGC |

\* Forward and Reverse-1 or -2 were used for sequencing of target region by sgADAR1-6 or sgADAR1-4 respectively.

effect on *RNASE L* KO cells with intact ADAR1. The kinetics of cell death or lack thereof can be observed in the movies shown in *Videos 1–4*.

To further demonstrate that endogenous dsRNA leads to RNase L activation thus accelerating cell death and that RNase L precludes recovery of *ADAR1* KO cells, we constructed an A549 cell line expressing murine coronavirus accessory protein NS2 (NS2-WT), which has a 2′,5′-phosphodiesterase (PDE) activity that cleaves 2-5A thereby antagonizing activation of RNase L. As a control, we also constructed a cell line expressing an inactive mutant NS2$^{H126R}$, unable to antagonize RNase L due to loss of its ability to cleave 2-5A (*Zhao et al., 2012*). Using CRISPR/Cas9 engineering (sgRNAs 4) we were able to recover *ADAR1* KO cells expressing functional NS2 PDE (NS2-*ADAR1* KO) but not the inactive mutant PDE (*Table 3*, *Figure 5a*) indicating that antagonism of RNase L by NS2 mediated cleavage of 2-5A as well as *RNASEL* KO promotes recovery of *ADAR1* KO cells. Transfection of NS2-*ADAR1* KO cells (clone C12) with siRNA against NS2 but not against control GFP reversed the NS2 phenotype, activated RNase L and promoted cell death by two days post-transfection while NS2-WT cells remained alive following siRNA mediated depletion of NS2 (*Figure 5b and c*). A second clone (C7) of NS2-*ADAR1* KO cells showed similar phenotype (*Figure 5—figure supplement 1*).

## Endogenous dsRNA leads to RNase L activation in *ADAR1* KO cells

We have shown that RNase L expression rescues ADAR1 KO cells and ectopic expression of active RNase L in *RNASEL-ADAR1* DKO cells or knockdown of the NS2 RNase L antagonist in NS2 expressing *ADAR1* KO cells leads to cell death. These findings suggest that in the absence of ADAR1 expression, OAS is activated by endogenous dsRNA leading to RNase L activation, thus impairing cell proliferation and survival. In order to directly demonstrate OAS activation and RNase L activity by endogenous dsRNA, we quantified 2-5A, the product of OAS, and RNA degradation by RNase L in several types of *ADAR1* KO cells in the absence of exogenously added dsRNA. For these experiments we used the *RNASEL-ADAR1* and *MAVS-ADAR1* DKO cells described above. In addition, for some of the assays we used *ADAR1* p150 KO and *MAVS-ADAR* p150 DKO cells, derived using CRISPR/Cas9 editing and sgRNA6 (*Table 3*; *Figure 2—figure supplement 1*). Both of the these cell lines are deficient in expression of the *ADAR1* p150 isoform (*Figure 2a*) previously reported in the mouse system to be responsible for the editing of RNA that activates MAVS or MDA5 (*Pestal et al., 2015*).

Because basal levels of 2-5A are minimal in the absence of exogenously added dsRNA, and the consequent RNase L activity very low (*Knight et al., 1980*) we pre-treated cells with IFN to increase OAS protein levels prior to quantifying 2-5A accumulation and RNA degradation. Indeed IFN treatment of ADAR1 KO cells upregulated ISG expression and further induced IFN by endogenous dsRNA in cells lacking expression of both isoforms of ADAR1 (p110 and p150) or ADAR1 p150 only (*Figure 6*). This was most evident in *RNASEL-ADAR1* DKO cells, which retained MAVS. Using an indirect FRET based assay (*Thakur et al., 2005*) we observed IFN-induced accumulation of 2-5A in *RNASEL-ADAR1* DKO and *MAVS-ADAR1* DKO cells (although the latter not to a statistically significant

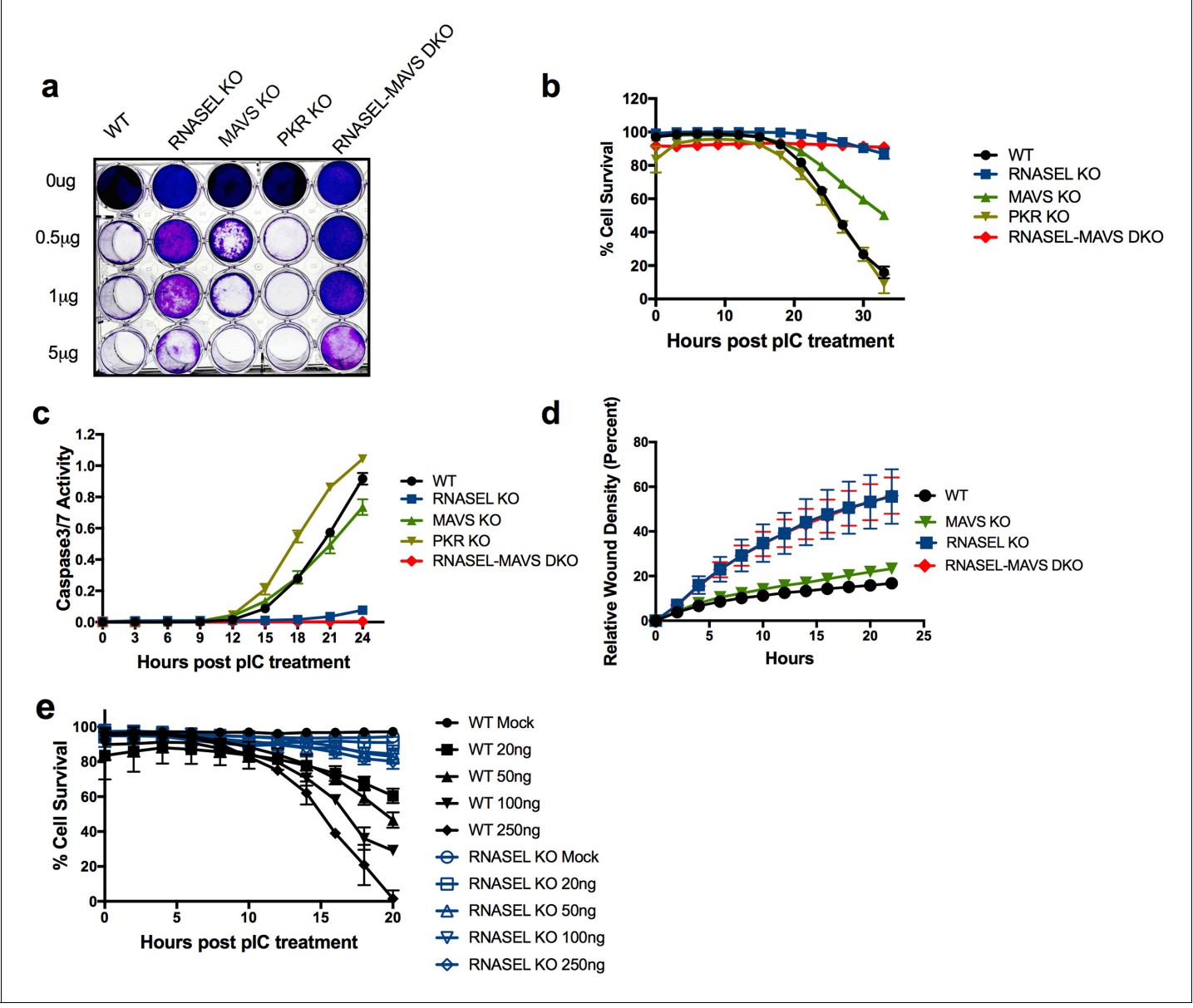

**Figure 3.** Ablation of RNase L activity attenuates pIC induced cell death and accelerates cell migration. (a) WT and *RNase L, MAVS, PKR* KO and *RNASEL-MAVS* DKO A549 cells were treated with pIC (0 to 5 µg/ml as indicated) and 48 hr later fixed and stained with crystal violet. Data are from one representative experiment of three. (b) WT and KO A549 cells as indicated were transfected with 20 ng/ml pIC and assessed for dead cells and total cells using an IncuCyte system calculated as describe in Methods. Four separate wells were treated with each experimental condition and a minimum of 4 image fields (>10000 cells per well) were analyzed per well. (c) Cells were transfected with 20 ng/ml of pIC and caspase-3/7 activity was determined using an IncuCyte system. Four separate wells were treated with each experimental condition and a minimum of 4 image fields (>10000 cells per well) were analyzed per well. Data represent the means±SD from a minimum of four independent replicates. Similar data were obtained from two additional independent experiments. (d) Cell monolayers were scratched and wound healing was assayed in the IncuCyte system. Wound closure was observed every hour at the indicated times by comparing the mean relative wound density of at least eight technical replicates and a minimum of 2 image fields (>1000 cells per well total) were analyzed per well. Error bars represent standard deviation (SD) from the mean of a minimum of eight. Similar data were obtained from two independent experiments. (e) WT or *RNASEL* KO HME cells were transfected with 0(mock) or 20, 50, 100 or 250 ng/ml pIC and cell viability was quantified in the IncuCyte system. Four biological replicate wells were treated with each experimental condition and a minimum of 4 image fields (>1000 cells per well total) were analyzed per well. Data is representative of four replicates.

The following source data is available for figure 3:

**Source data 1.** Excel data for *Figure 3*.

**Table 3.** Knockout of ADAR1 or ADAR1p150 from WT, *MAVS* KO, *RNASEL* KO, *PKR* KO, NS2-WT or NS2^H126R –WT cells.

| Genotype of parent cells | sgRNA | Resulting KO cells | | |
|---|---|---|---|---|
| | | Name | Number of clones screened | Number of KO clones obtained |
| WT | sgADAR1-6 | p150 KO | 24 | 12 |
| | sgADAR1-4 | NA* | 17 | 0 |
| *MAVS* KO | sgADAR1-6 | *MAVS*-p150 KO | 24 | 8 |
| | sgADAR1-4 | *MAVS-ADAR1* DKO | 23 | 8 |
| *RNASEL* KO | sgADAR1-4 | *RNASEL-ADAR1* DKO | 24 | 2 |
| *PKR* KO | sgADAR1-4 | NA | 24 | 0 |
| WT NS2 | sgADAR1-4 | NS2-*ADAR1* KO | 56 | 4 |
| WT NS2^H126R | sgADAR1-4 | NA | 54 | 0 |

*NA, Not available.

extent) but not in WT, *RNASEL* KO or *MAVS* KO cells (*Figure 7a*). Furthermore, while rRNA remained intact in WT, *RNASEL* KO or *MAVS* KO cells, highly specific and characteristic RNase L-mediated rRNA degradation products (*Silverman et al., 1983*) identical to the pattern induced by Sindbis infection of WT A549 (*Figure 7b*) were observed by Bioanaylzer in IFN treated ADAR1 p150 KO, *MAVS-ADAR1 p150* DKO, or *MAVS-ADAR1* DKO cells. (It is important to note that the RNA degradation products observed in IFN-treated *RNASE L-ADAR1* DKO cells are not the characteristic specific RNase L mediated pattern. The nuclease(s) responsible for these non-specific rRNA cleavage products is unknown.) Thus 2-5A was produced and RNase L activated by endogenous RNA even in the absence of MAVS. In addition to cleaving rRNA, RNase L was recently reported to strongly and site-specifically cleave tRNA-His at position 36 located in the anti-codon loop. The site-specific tRNA-His cleavage was readily observed using a quantitative RT-PCR assay in IFN-treated *MAVS-ADAR1* DKO, but not in IFN-treated cells, which lack RNase L activity in the above assays: *MAVS* KO and *RNASEL-ADAR1* DKO cells (*Figure 7c*) (*Donovan et al., 2016*).

## Discussion

Ablation of ADAR1 was previously shown to be lethal during mouse embryogenesis (*Pestal et al., 2015*; *Liddicoat et al., 2015*; *Mannion et al., 2014*). *Adar1* KO embryos show increased levels of IFN production and ISG expression relative to WT and die by E12.5, implicating ADAR1 editing of dsRNA in regulating cell and animal survival. However, deletion of MDA5 or MAVS, components of the canonical type I IFN production pathway (*Figure 1*), was shown to overcome the ADAR1 deficiency and allow the viability of mouse embryos, although most of the surviving mice died within the first few days after birth and there were abnormalities in organ development as well as excess IFN signaling (*Pestal et al., 2015*). MDA5/MAVS dependent IFN production and signaling involves many pathways and IFN stimulated gene products and it remained unclear which of the these are primary contributors to the death of these embryos, although it has been shown that PKR ablation was unable to rescue either *Adar1* KO mouse embryo lethality (*Wang et al., 2004*).

While our finding that MAVS KO rescued the lethality of ADAR1 in a human cell system is consistent with the literature on *Adar1* KO in murine systems in vivo and expected, the finding that RNase L ablation rescues the ADAR1 lethality even in the presence of MAVS and normal IFN response (*Figure 2*; *Table 3*) is unexpected. It is possible that, in the mouse embryo system, RNase L, activated downstream of MDA5-MAVS also contributes to embryonic lethality. However, rescue of lethality due to *Adar1* KO by *Mda5* or *Mavs* KO in the mouse embryo may be due to one or more other MAVS dependent activities. It should be noted that mouse embryo fibroblasts (mef) (*Mannion et al., 2014*) and HEK293T cells (*Pestal et al., 2015*) can survive *ADAR1* KO. This is likely due to the low levels of OAS expression in these cell types (*Banerjee et al., 2014*) (and data not shown) and in the case of mefs, also a low level of RNase L (*Banerjee et al., 2014*).

The observation that ectopic expression of active RNase L, but not mutant inactive RNase L, promotes death of ADAR1 deficient cells supports the central role of RNase L activation in dsRNA-

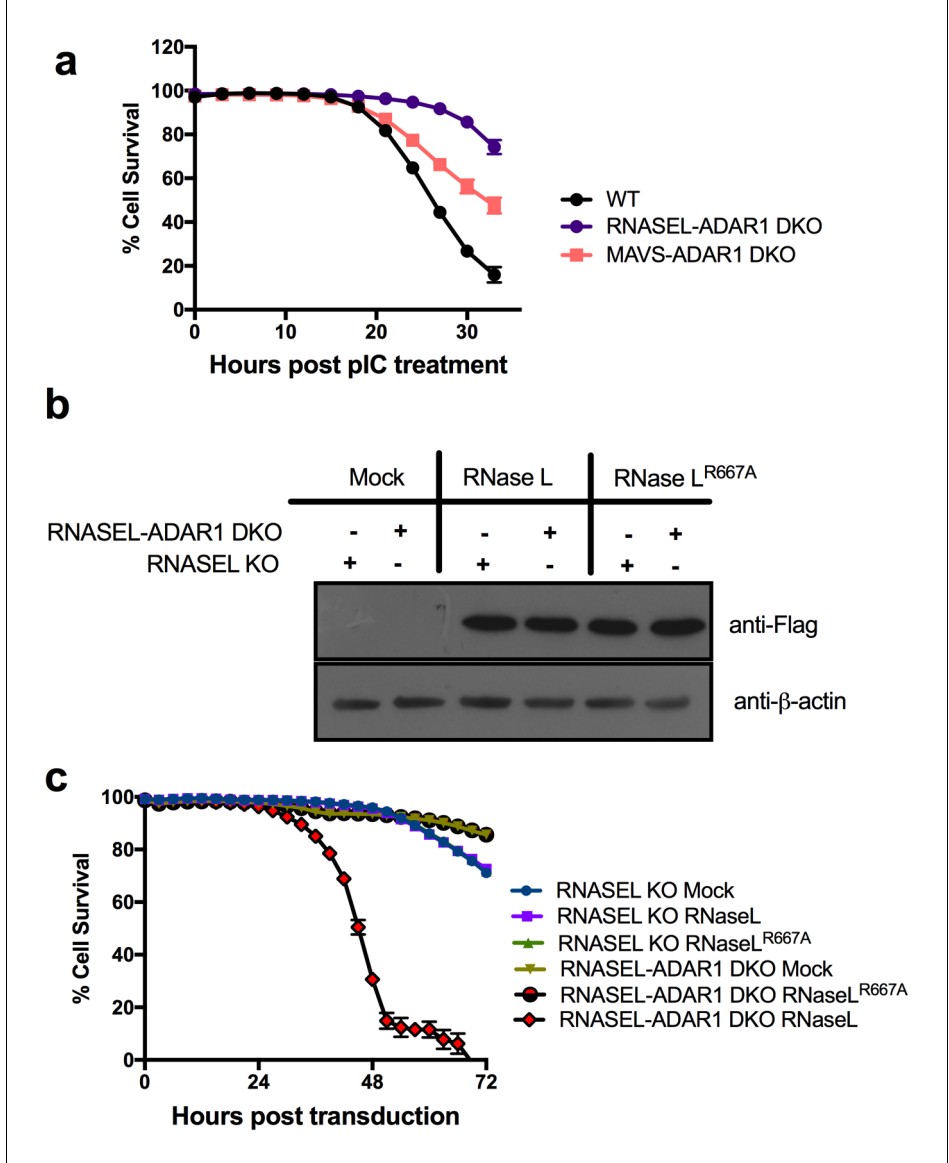

**Figure 4.** Expression of WT RNase L promotes death of *RNASEL-ADAR1* KO cells. (a) WT and KO A549 cells as indicated were transfected with 20 ng/ml of pIC and assessed for dead cells and total cells. Four separate wells were treated with each experimental condition and a minimum of 4 image fields (>10000 cells per well) were analyzed per well. Data represent the means SD from a minimum of four independent replicates. Similar data were obtained from two additional independent experiments. (The data for WT cells are the same as shown in *Figure 3b*). (b) *RNASEL* KO or *RNASEL-ADAR1* DKO cells transduced with lentiviruses expressing either WT RNase L or inactive mutant (R667A) RNase L, lysed and proteins analyzed by Western immunoblotting with anti-Flag M2 monoclonal antibody (upper) or with anti-β-actin antibody (lower). (c) *RNASEL* KO or *RNASE L-ADAR1* DKO cells were transduced with lentiviruses expressing either WT RNase L or inactive mutant (R667A) RNase L and dead and total cells assessed in the IncuCyte system. A minimum of four separate wells was treated with each experimental condition and a minimum of 9 image fields were analyzed per well. Data represent the means SD and are from one of two representative experiments. Movies showing cells from 0–72 hr post transduction are shown in *Videos 1–4*. See *Figure 4—source data 1*.

The following source data is available for figure 4:

**Source data 1.** Excel data for *Figure 4*.

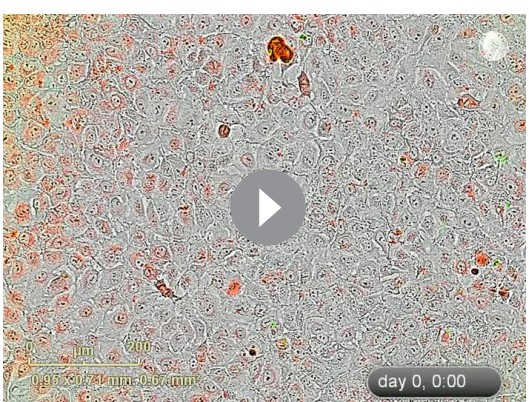

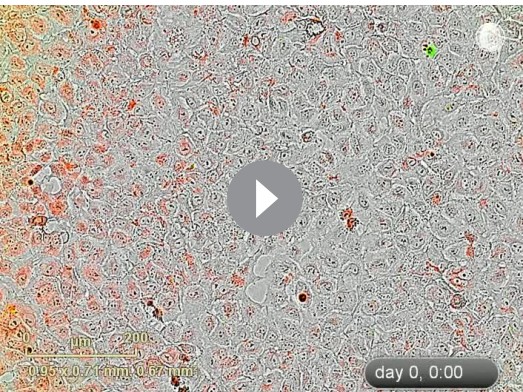

**Video 1.** *RNASEL* KO cells from 0–72 hr post transduction with lenti-RNASE L (as in *Figure 4c*).

**Video 2.** *RNASEL* KO cells from 0–72 hr post transduction with lenti-RNase L^R667A (as in *Figure 4c*).

mediated cell death (*Figure 4c*, *Videos 1–4*). Further supporting this conclusion, expression of the MHV PDE, NS2, a potent RNase L antagonist (*Zhao et al., 2012*), but not its inactive mutant, enables recovery of viable *ADAR1* KO cells while knockdown of NS2 in *ADAR1* KO cells leads to cell death (*Figure 5*). Based on these results, we propose that RNase L has a predominant role in promoting cell death by dsRNA (*Figure 1*). It has been shown recently that RNase L site-specifically cleaves tRNAs and Y-RNAs, and that depletion of the Y-RNA RNY1 contributes to apoptosis (*Donovan et al., 2016*).

In contrast to RNase L, and consistent with mouse embryo studies (*Wang et al., 2004*), deletion of PKR did not allow recovery of *ADAR1* KO cells and single PKR KO did not change the kinetics of pIC-induced death in cells that contained ADAR1 (*Table 3*, *Figure 3*). Apparently, PKR (a dsRNA-activated protein kinase that inhibits translation) does not substantially contribute to ADAR1 KO cell lethality. [PKR was activated by pIC in WT A549 cells resulting in phosphorylation of eIF2$\alpha$, whereas pIC treatment of PKR KO cells did not result in phosphorylated eIF2$\alpha$, (*Figure 2d*).] However, it is noted that IFN treatment of ADAR1 deficient cells did not induce PKR expression, in contrast to ADAR1 expressing cells (*Figure 2a*, *Figure 5a*, *Figure 2—source data 2–source data1*; *Figure 5—source data 5–source data1*). We do not currently know why this occurs.

Endogenous cellular dsRNA activates OAS leading to RNase L activity in *MAVS-ADAR1* KO cells as evidenced by rRNA degradation, site-specific tRNA-His cleavage following IFN stimulation, and accumulation of 2-5A in *RNASEL* KO cells, even in the absence of exogenous dsRNA (*Figure 7*). It should be noted that dsRNA is the only well-established activator of OAS enzymatic activity [

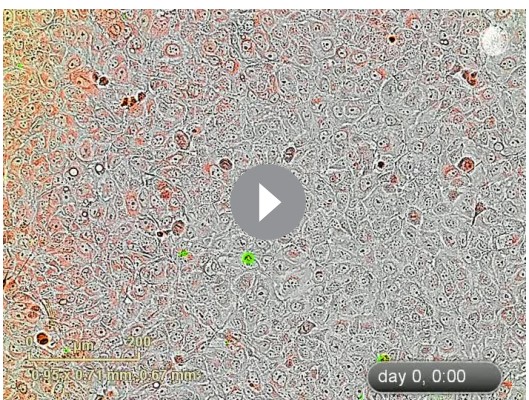

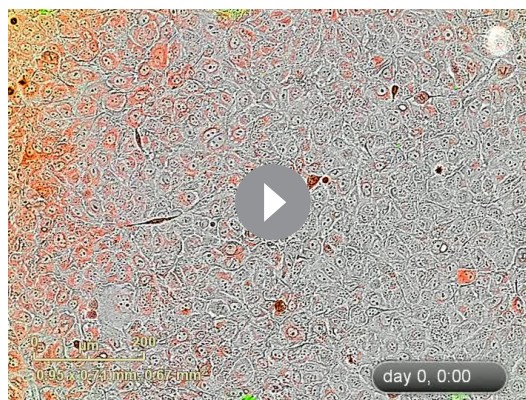

**Video 3.** *RNASEL-ADAR1* DKO cells from 0–72 hr post transduction with lenti-RNASE L (as in *Figure 4c*).

**Video 4.** *RNASEL-ADAR1* DKO cells from 0–72 hr post transduction with lenti-RNase L^R667A (as in *Figure 4c*).

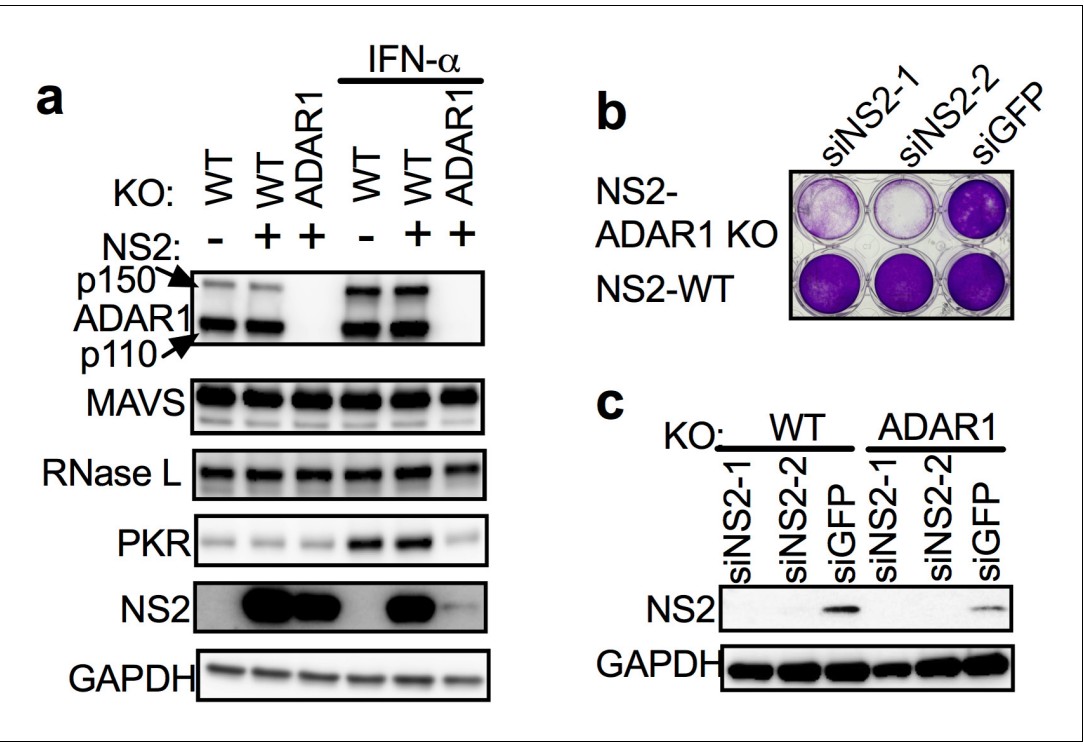

**Figure 5.** Expression of the MHV NS2 RNase L antagonist activity can rescue *ADAR1* KO A549 cells. (**a**) WT A549, WT A549 NS2 expressing cells or *ADAR1* KO NS2 (clone C12) cells were mock treated or treated with 1000 U/ml of IFN-α overnight, lysed and proteins analyzed by Western immunoblotting with antibodies as indicated (PKR expression was induced 3.5–4.0 fold following IFN treatment in cells expressing ADAR1 and reduced 0.6 fold in cells deleted for ADAR1; *Figure 5—source data 1*.) (b and c) *ADAR1* KO or WT A549 NS2 expressing cells were transfected with siRNAs against NS2 (siNS2-1 and siNS2-2) or siRNA against GFP (siGFP) and (**b**) 96 hr later cells were fixed and stained with crystal violet or (**c**) 72 hr later lysed and analyzed by Western immunoblotting. The siRNA knockdown data (**b**) is from one representative experiment of three. Similar data were obtained with another *ADAR1* KO NS2 clone (C7) as shown in *Figure 5—figure supplement 1*.

The following source data and figure supplement are available for figure 5:

**Source data 1.** Quantification of PKR induction (+/− IFN) treatment for *Figure 5a*
**Figure supplement 1.** Knock down of NS2 expression in an additional clone of *ADAR1* KO NS2 expressing cells causes cell death.

---

*Silverman and Weiss, 2014* and references therein]. While IFN treatment of *RNASEL-ADAR1* DKO cells produces enhanced expression of IFN-β and -λ, and ISGs, these effects of IFN were reduced in *MAVS-ADAR1* DKO cells. Our findings suggest that MAVS is likely to augment RNase L activation through the upregulation of IFNs and OASs (which are ISGs), effects that do not require exogenous dsRNA when ADAR1 is deficient (*Figure 6*). Indeed, although not statistically significant, the accumulation of 2-5A was greater in *MAVS-ADAR1* DKO cells compared to *MAVS* (*Figure 7*). Moreover, deletion of MAVS promotes the viability of A549 cells engineered to lack ADAR1 and upon pIC treatment, the *MAVS* KO cells exhibit a slower rate of cell death than WT A549 (*Figure 3*). Nevertheless, *MAVS-ADAR1* DKO cells die faster than *RNASEL-ADAR1* DKO cells most likely because RNase L can be activated in the former cells (*Figure 4a*; *Figure 7b and c*). Thus, while the MDA5/MAVS pathway plays a central role in IFN induction and signaling and can upregulate both OAS and PKR (*Figure 1*), these data indicate that RNase L can be activated in the absence of MAVS expression in *MAVS-ADAR1* DKO cells by pIC. These results are consistent with our previous findings that activation of RNase L during virus infections does not depend on IFN induction (*Birdwell et al., 2016*; *Li et al., 2016*).

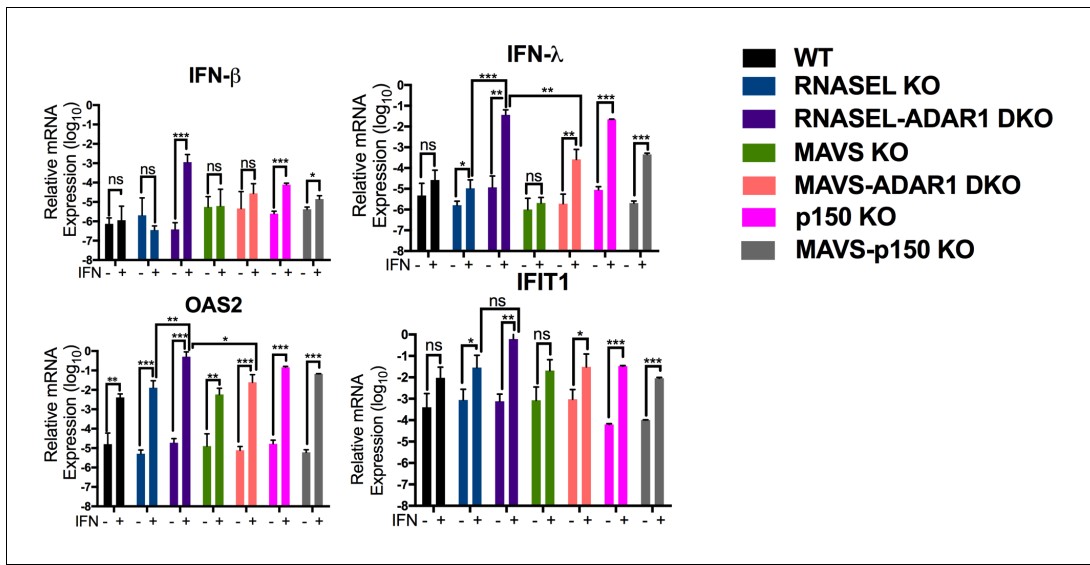

**Figure 6.** Interferon (IFN) and interferon stimulated gene (ISG) expression following IFN treatment of *ADAR1* KO cells. Cells were treated or mock treated with 10 U/ml of IFN-α and 24 hr post treatment, lysed and RNA extracted. Relative mRNA levels for (a) IFN-β, (b) IFN-λ, (c) OAS2 and (d) IFIT1 were quantified by RT-qPCR, using primers listed in **Table 4**, and expressed as $2^{-\Delta CT}$ where $\Delta C_T = C_{T(gene\ of\ interest)} - C_{T(\beta-actin)}$. The data are the average of three independent biological replicates (each with three technical replicates) expressed as means ± SD; *p<0.05, **p<0.01, ***p<0.001. See **Figure 6—source data 1** (includes exact P values).

The following source data is available for figure 6:

**Source data 1.** Excel data for **Figure 6**.

Since RNase L is activated in the absence of ADAR1 editing contributing to cell death, RNase L-induced cell death could potentially contribute to symptoms of the neurodegenerative and inflammatory disease Aicardi-Goutiéres syndrome (AGS) in cases with mutation of *ADAR1* (*Rice et al., 2012*). In mouse models, mutations of genes responsible for AGS, such as RNase H2 (*Pokatayev et al., 2016*) and Trex1 (*Gray et al., 2015*), led to self-DNA-mediated activation of cGAS, a protein functionally and structurally similar to OASs (*Hornung et al., 2014*). Our findings show that mutation of ADAR1 leads to self-dsRNA activation of one or more OAS isoform resulting in RNase L activity and cell death. We suggest that mutations affecting homologous nucleic acid sensors in the IFN system, cGAS and OAS, both of which produce unusual 2'−5' internucleotide linkages, lead to AGS although through different signaling pathways. In vivo evidence for the pro-inflammatory role of RNase L was obtained in virus-infected mice where presence of RNase L contributed to secretion of the pro-inflammatory cytokine IL-1β (*Chakrabarti et al., 2015*). Therefore, an RNase L inhibitor drug could potentially reduce tissue damage and alleviate inflammation in some circumstances.

## Materials and methods

### Cell lines and viruses

Human A549 cells, obtained from Dr. George Stark (Cleveland Clinic) authenticated by STR analysis by ATCC (compared to ATCC CCL-185) were cultured in RPMI 1640 medium (Gibco) supplemented with 10% fetal bovine serum (FBS), 100 U/ml penicillin and 100 μg/ml streptomycin. Human HEK 293 T cells, authenticated by STR analysis by ATCC (compared to ATCC CRL-1573), were cultured in DMEM supplemented with 10% FBS and 1 mM sodium pyruvate. hTERT-HME1 cells (HME), obtained from Clonetech, authenticated by STR analysis at Genetica DNA Labs (compared to ATCC CRL-4010), were cultured with mammary epithelial cell growth medium (MEGM, Lonza CC-3150). Cell

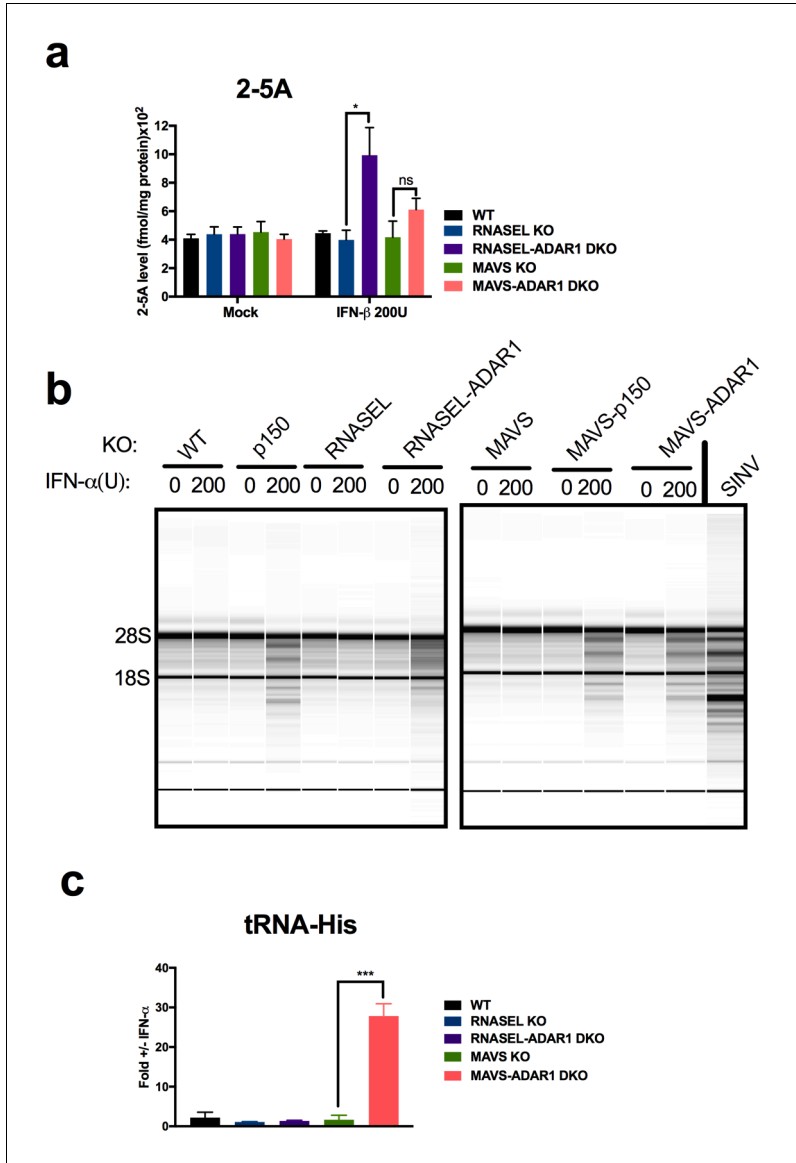

**Figure 7.** RNase L activation following IFN treatment of *ADAR1* KO cells. (a) Cells were mock treated or treated with 200 U/ml of IFN- and cells were lysed and 2-5A quantified using a FRET based assay. The data are the average of three independent biological replicates, expressed as means SD, **p=0.014; ns, p=0.1. (b) Cells were treated with 200 U/ml of IFN-α or infected with Sindbis virus (SINV) *(Frolova et al., 2002)* at MOI = 1, and at 48 hr post treatment or 24 hr post infection, cells were harvested, total RNA was extracted and resolved on RNA chips on a Bioanalyzer. The position of 28S and 18S rRNA and indicated. Data shown are from one representative experiment of two. (c) Cells were treated or mock treated with 10 U/ml of IFN-α and 24 hr post treatment and lysed. RNA was extracted and specific cleavages at tRNA-His-36 were quantified. The data are the average of three independent biological replicates, expressed mean SD; ***p=0.00019. See *Figure 7—source data 1*.

The following source data is available for figure 7:

**Source data 1.** Excel data for *Figure 7*.

lines tested negative for mycoplasma at University of Pennsylvania (A549 or 293 T cells) or the Cleveland Clinic (HME cells). The RNase L knockout A549 cells were constructed using CRISPR-cas9 gene editing technology as we described previously (*Li et al., 2016*).

Table 4. qRT-PCR primers for human *actin, IFN, OAS2* and *IFIT1* genes.

| Genes | Forward (5'–3') | Reverse (5'–3') |
|---|---|---|
| ACTIN | ACTGGAACGGTGAAGGTGAC | GTGGACTTGGGAGAGGACTG |
| IFN-β | GTCAGAGTGGAAATCCTAAG | ACAGCATCTGCTGGTTGAAG |
| IFN-λ | CGCCTTGGAAGAGTCACTCA | GAAGCCTCAGGTCCCAATTC |
| OAS2 | TTCTGCCTGCACCACTCTTCAACGA | GCCAGTCTTCAGAGCTGTGCCTTTG |
| IFIT1 | TGGTGACCTGGGGCAACTTT | AGGCCTTGGCCCGTTCATAA |

## Antibodies

Rabbit anti-ADAR1 D7E2M antibody (1:1000, Cell Signaling), mouse monoclonal anti-MAVS E3 (1:250, Santa Cruz), mouse monoclonal anti-RNase L (1:1000) (*Dong and Silverman, 1995*), mouse monoclonal anti-PKR B-10 antibody (1:200, Santa Cruz), mouse monoclonal anti-GAPDH GA1R (1:1000, Thermo-Fisher), mouse monoclonal anti-NS2 (1:600) (provided by Stuart Siddell, University of Bristol, United Kingdom (*Gusho et al., 2014*) were used to detect ADAR1, MAVS, RNase L, PKR, GAPDH and NS2 respectively. Mouse monoclonal anti-Flag M2-antibody (1:5000) and mouse mono-clonal anti-ß-actin (1:50000) were from Sigma-Aldrich. Rabbit anti-eIF2α (1:1000) or anti-Ser-51-phosphorylated eIF2α (1:1000) were from Cell Signaling Technology. Secondary antibodies were goat anti-mouse antibody (1:5000, Santa Cruz) and goat anti-rabbit antibody (1:3000, Cell Signaling) conjugated with HRP(Horseradish peroxidase).

## Construction of plasmids and pseudolentivirus

The oligonucleotide sequences to be used for generation of single guide RNAs (sgRNA) are listed in *Table 1* and their positions of sgRNA4 and sgRNA6 in the ADAR1 gene are shown in *Figure 2—figure supplement 1*. A pair of forward and reverse oligonucleotides for generation of each sgRNA (*Table 1*), synthesized by IDT) were annealed by published methods (*Ran et al., 2013*) and were inserted into plasmid vectors pLenti-CRISPR(Addgene) between BsmBI restriction sites. The resulting plasmids were pLenti-sgADAR1-4 and pLenti-sgADAR1-6 (targeting the ADAR1 gene), pLenti-sgMAVS-1(targeting the *MAVS* gene) pLenti-sgPKR-1 (targeting the *PKR* gene). The cDNAs encoding murine coronavirus mouse hepatitis virus (MHV) NS2 and mutant NS2$^{H126R}$ were cloned into a pLenti vectors using SalI/XhoI restriction sites. The packaging of pseudo lentiviruses was described previously (*Li et al., 2016*).

## Construction of MAVS, PKR, ADAR1 gene knockout cells

The methods for construction of single gene knockout (KO) A549 cells using Lenti-CRIPSR were described previously (*Li et al., 2016*). For the double gene knockout (DKO), *RNaseL*-KO or *MAVS*-KO cells were transduced with Lenti-sgADAR1-4 to generate *RNaseL-ADAR1* DKO or *MAVS/ADAR1* DKO cells. For knocking out *ADAR1* p150 expression, A549 or *MAVS*-KO cells were transduced with Lenti-sgADAR1-6, the cells were selected for puromycin resistance and cloned by previous described methods (*Li et al., 2016*). Mutations in the cells were verified by PCR using primers listed in *Table 2*.

## Construction of NS2 or NS2$^{H126R}$ expressing cells and NS2 knockdown

A549 cells were transduced with Lenti-NS2 or Lenti-NS2$^{H126R}$ and 48 hr post transduction, medium containing 1 mg/ml of hygromycin was added, after 72 hr selection the cells were plated into a new flask and 24 hr later cloned by limited dilution by previous methods (*Li et al., 2016*). For NS2 knockdown, $3 \times 10^5$ cells were plated in one well of a 24-well plate, and transfected with 100 pmol siRNA in Lipofectatimine 2000; cells were harvested for western blotting (72 hr post transfection) to determine the expression level of NS2 or fixed with 4% PFA (96 hr post transfection) and stained with crystal violet. siNS2-1 (5' rArUrGrArArGrArUrArUrUrGrUrUrGrArArArUrUrCrUrCCG 3') and siNS2-2 (5' rGrGrArArUrGrCrArArUrUrCrUrArUrCrArUrArArArGrAAC 3') targetting NS2 were designed and synthesized by IDT and the siGFP was purchased from SantaCruz.

## Western immunoblotting

Confluent cells in 12-well plates were treated or mock treated with 1000 U of human IFN-α (PBL) overnight, harvested, washed in PBS and lysed with NP40 buffer with protease inhibitor cocktail (Roche). Cell lysates, containing equal amounts of protein, were mixed with 4X Laemmli buffer and boiled at 95°C for 5 min and analyzed by electrophoresis on 4–15% gradient SDS gels. Proteins were transferred to polyvinylidene difluoride membranes, which were treated with 5% nonfat milk in TBST (Tris-HCl buffer saline with 0.5% Tween-20) blocking buffer for 1 hr, followed by incubation overnight at 4°C with antibodies diluted into TBST. Membranes were then washed three times with TBST and incubated with secondary antibodies for 1 hr at room temperature, washed three times with TBST and then incubated with SuperSignal West Dura Extended Duration substrate (Thermo Fisher) and the signal was detected using an Amersham Imager 600(GE). The expression level of PKR (*Figure 2a* and *Figure 5a*) was quantified by ImageJ software and the fold change between IFN-α treated and mock treated samples was calculated.

## Crystal violet cell death assay

Cells were transfected with pIC (0.4 kb, Millipore), amounts as indicated, as previously described (*Li et al., 2016*). At 48 hr post transfection cells were fix with 4% paraformaldehyde and stained with 1% crystal violet.

## IncuCyte system assays
### Cell death Assay

Cells ($1 \times 10^5$) were seeded in 24-well plates and after 12 hr were transfected with 20 ng/ml of pIC (high molecular weight (1.5–8 kb, InvivoGen) with Lipofectamine 2000 (Invitrogen) according to manufacturer's protocol. The cells were incubated with 250 nM Sytox-Green dye (Thermo Fisher), a nucleic acid stain that is an indicator of dead cells and which is impermeant to live cells, and 250 nM of cell permeable dye Syto 60-Red (ThermoFisher), which allows quantification of the total number of cells present in each field, using an IncuCyte Live-Cell Imaging System and software (Essen Instruments 2015A) for 36 hr. Cell death was measured by counting the green objects per mm$^2$ (dead cells, green) and then normalizing to the total number of cells per mm$^2$ (red objects) at each time point using IncuCyte software. The WT and *RNASEL* KO HME ($2 \times 10^5$) cells were grown in 12-well plates and transfected with different concentration of pIC (250 ng/ml, 100 ng/ml, 50 ng/ml and 20 ng/ml). Cell death was measured with the IncuCyte system with Sytox-Green dye (250 nM) for 20 hr.

## Apoptosis assay (caspase 3/7)

Cells were seeded and transfected with pIC as above. The cells were incubated with IncuCyte Kinetic Caspase-3/7 Apoptosis Assay Reagent (green) (Essen Bioscience) according to manufacturer's protocol and with 250 nM of Syto Red dye (ThermoFisher) for 24 hr. The apoptotic index was determined by counting the green object count/mm$^2$ and then normalized by red object count/mm$^2$ using IncuCyte software.

## Cell migration assay

The wound healing assays were done by seeding cells in 96-well plates (Essen Image-Lock plates, Essen Instruments), and incubating in serum-free medium for 24 hr as described previously (*Banerjee et al., 2015*). The wounds were made with a wound scratcher (Essen Instruments). Cells were stimulated with 10% serum immediately after wound scratching, and wound confluence was monitored with IncuCyte Live-Cell Imaging System and software (Essen Instruments 2015 A). Wound closure was observed every hour at the indicated times by comparing the mean relative wound density of at least eight biological replicates in each experiments.

## Re-constitution of RNase L KO cells with lentivirus encoding flag RNase L (WT and R667A mutant)

The 3X FLAG-human RNase L DNA or human mutant 3X FLAG-RNase L R667A DNA (*Chakrabarti et al., 2015*) were PCR amplified and inserted in lentiviral vector pCMV-PL4-Neo (Addgene, Principal Investigator Eric Campeau) as described previously (*Chakrabarti et al., 2015*; *Banerjee et al., 2015*). The lentiviruses were produced in HEK293T cells as described

(*Banerjee et al., 2015*). Virus-containing medium was collected 48 hr after transfection and stored at −80°C. ADAR1-RNase L DKO or RNase L KO cells were infected at a density of $10^5$ cells ml$^{-1}$ in the presence of 8 μg ml$^{-1}$ polybrene, in 24-well plates. Cell survival was monitored with the IncuCyte System by staining with Sytox-Green and Syto-Red as described above.

## Quantification of 2-5A

Intracellular 2-5A was quantified by an indirect assay in which activation of RNase L activity is measured (*Elbahesh et al., 2011*). In brief, IFN treated (200 U/ml human IFN-β) for 48 hr or mock treated cells were washed with PBS, and lysed in preheated (95°C) Nonidet P-40 lysis buffer (50 mM Tris-HCl, pH 7.2, 0.15 M NaCl, 1% Nonidet P-40, 200 μM sodium orthovanadate, 2 mM EDTA, 5 mM MgCl$_2$, 5 mM DTT) and heated to 95°C for another 7 min. The cell extracts were centrifuged for 10 min at 14,000 × g at room temperature, and the cleared supernatants collected. Levels of 2-5A were determined by RNase L-based FRET assays in comparison to a standard curve of authentic (2′−5′) p$_3$A$_3$ as described earlier (*Thakur et al., 2005*). Briefly, recombinant human RNase L was incubated with purified authentic (2′−5′)p$_3$A$_3$ or cell extract (normalized for protein concentration prior to centrifugation). The FRET RNA probe was added and the reaction mixtures incubated at 20°C for 2 hr. Fluorescence was measured with a Wallac 1420 fluorimeter (Perkin-Elmer) as described previously (*Thakur et al., 2005*).

## Ribosomal RNA (rRNA) cleavage assay

Cells were treated with 200 U/ml of IFN-α or infected with Sindbis virus (SINV) (*Frolova et al., 2002*) at MOI = 1, and at 48 hr post treatment or 24 hr post infection, cells were harvested in RLT cell lysis buffer (RNeasy mini kit, Qiagen). Total RNA was extracted and was resolved on RNA chips using an Agilent 2100 BioAnalyzer to determine the integrity of 28S and 18S rRNA.

## RtcB-ligase assisted qPCR

RNA fragments generated by RNase L were detected as described (*Donovan et al., 2016*). Briefly, cells were treated with 10U of IFN-α and 24 hr post treatment, total RNA was purified by Trizol and RNAs with 2′−3′ cyclic phosphate were ligated to the adapter 5′ rGrArUrCrGrUCGGACTGTAGAAC TCTGAAC 3′ using RtcB RNA ligase. EDTA-quenched ligation reactions were reverse transcribed using Multiscribe reverse transcriptase (ThermoFisher) and the primer 5′ TCCCTATCAGTGATAGA-GAGTTCAGAGTTCTACAGTCCG 3′. The resulting cDNAs were assessed for RNase L cleavage products by qPCR for specific cleavage sites and normalized to U6, which has a naturally occurring 2′−3′ cyclic phosphate. The qPCR primers for tRNA-His-36 cleavage are 5′ GTTAGTACTCTGCGTTG TGGA 3′ (Forward) and 5′ TCCCTATCAGTGATAGAGAG 3′ (reverse). U6 primers for normalization are 5′ GCTTCGGCAGCACATATACTA 3′ (forward) and 5′ CGAATTTGCGTGTCATCCTTG 3′ (reverse).

## mRNA quantification by quantitative reverse transcriptase-PCR (qRT-PCR)

Cells were treated with 10 U/ml of IFN-α (PBL) and 24 hr post treatment, cells were lysed in RLT Plus RNA lysis buffer (Qiagen), and RNA was isolated using the RNeasy Plus Mini Kit (Qiagen) as previously described (*Zhao et al., 2012*). Relative IFN-α, IFN-λ, OAS2, IFIT2 mRNA expression levels were quantified as $\Delta C_T$ values relative to actin mRNA [$\Delta C_T = C_{T(gene\ of\ interest)} - C_{T(\beta\text{-}actin)}$] and expressed using the formula $2^{-\Delta CT}$. Primer sequences are listed in *Table 4*.

## eIF2α phosphorylation

WT and PKR-KO A549 cells were seeded at a density $5 \times 10^5$ cells/well of in six-well plates. After 16 hr the cells were transfected with 1 μg/ml pIC using Lipofectamine 2000 and incubated for 2 hr. The cells were harvested in NP-40 lysis buffer as above. The lysates were centrifuged at 1000Xg for 12 min at 4°C. The total protein amounts in the cleared supernatants were determined by the Bradford method (Bio-Rad), and 30 μg of total protein was separated on 12% SDS-PAGE, transferred onto polyvinylidene difluoride(PVDF)membranes (0.45 μm) (BioRad), and probed with either antibody against total anti-eIF2α or Ser-51-phosphorylated eIF2α(Cell signaling Technology).

## Statistics

Plotting of data was performed using GraphPad Prism software (GraphPad Software, Inc., CA). Statistical analysis were performed using Excel software (Microsoft Office). Statistical significance was determined by an unpaired two-tailed Student t test.

## Acknowledgements

Research reported in this publication was supported by National Institute of Allergy and Infectious Disease, National Institute of Neurological Disease and Stroke and the National Cancer Institute, and National Institutes of General Medicine of the National Institutes of Health (NIH) under award numbers R01AI104887 (to SRW and RHS), R01-NS-080081 (SRW), R01CA044059 (to RHS) R01GM110161 (AK) Princeton University, Sidney Kimmel Foundation Grant AWD1004002 (AK), Burroughs Wellcome Foundation Grant 1013579 (AK) and Valle Foundation 23307 G0002-10009096 (AK). S.A.G. was supported in part by 4T32AI007324 and SR was supported in part by 5T32GM007388.

## Additional information

### Funding

| Funder | Grant reference number | Author |
| --- | --- | --- |
| National Institute of Allergy and Infectious Diseases | R01AI104887 | Robert H Silverman Susan R Weiss |
| National Institute of Neurological Disorders and Stroke | R01NS080081 | Susan R Weiss |
| National Cancer Institute | R01CA044059 | Robert H Silverman |
| Burroughs Wellcome Fund | Grant 1013579 | Alexei Korennykh |
| Sidney Kimmel Foundation for Cancer Research | AWD1004002 | Alexei Korennykh |
| National Institute of General Medical Sciences | R01GM110161 | Alexei Korennykh |
| Vallee Foundation | 23307-G0002-10009-96 | Alexei V Korennykh |
| National Institute of Allergy and Infectious Diseases | T32AI007324 | Stephen A Goldstein |
| National Institute of General Medical Sciences | T32GM007388 | Sneha Rath |

The funders had no role in study design, data collection and interpretation, or the decision to submit the work for publication.

### Author contributions

YL, Conceptualization, Data curation, Formal analysis, Validation, Investigation, Methodology, Writing—original draft, Project administration, Writing—review and editing; SB, Conceptualization, Data curation, Formal analysis, Supervision, Validation, Investigation, Methodology, Writing—original draft, Project administration, Writing—review and editing; SAG, Data curation, Formal analysis, Validation, Investigation, Writing—original draft, Writing—review and editing; BD, CG, Data curation; SR, Data curation, Software, Formal analysis, Supervision, Funding acquisition, Validation, Investigation, Methodology, Writing—original draft, Project administration, Writing—review and editing; JD, Conceptualization, Resources, Data curation, Validation, Investigation, Visualization, Methodology, Writing—original draft, Project administration, Writing—review and editing; AK, RHS, SRW, Conceptualization, Resources, Data curation, Formal analysis, Supervision, Funding acquisition, Validation, Investigation, Methodology, Writing—original draft, Project administration, Writing—review and editing

Author ORCIDs

Yize Li, http://orcid.org/0000-0003-1721-741X

Susan R Weiss, http://orcid.org/0000-0002-8155-4528

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
