## [Decision Letter]

Thank you for submitting your article "Ribonuclease L Mediates the Cell-Lethal Phenotype of Adenosine Deaminase 1 Deficiency in Human Cells" for consideration by *eLife*. Your article has been reviewed by three peer reviewers, and the evaluation has been overseen by a Reviewing Editor and James Manley as the Senior Editor. The following individual involved in review of your submission has agreed to reveal her identity: Brenda Bass (Reviewer #2).

The reviewers have discussed the reviews with one another and the Reviewing Editor has drafted this decision to help you prepare a revised submission.

We have now received comments on your manuscript from three experts in the specific area of ADAR1 biology. All agreed that the work was interesting, well done and in principle suitable for *eLife*. Nevertheless the following fairly minor points should be thoroughly addressed via revision.

1) Because ADAR1 technically belongs to the cytosine deaminase family of enzymes, a more appropriate title would be to substitute adenosine deaminase with the "double stranded RNA editing enzyme ADAR1 in a human cell line".

2) ADAR1 activity does not separate dsRNA into single stranded RNA but rather creates partially dsRNA. Please correct in the text and figure.

3) Please remove the word partially from "partially rescues" in the Abstract and also in the Introduction.

4) Indicate the source and size of poly IC used.

5) Please quantify the Western blots in Figure 2 and Figure 3 and comment on the levels of PKR seen in interferon treated cells.

6) The reviewers agreed that the conclusions derived from this work should be tempered somewhat because results obtained primarily in a single cell line cannot necessary be extrapolated to the intact organism. Because mouse data indicates that the RIG-I/MDA5 is perhaps the primary pathway you could modify your conclusion to indicate that the OAS/RNase L pathway is an important contributor at least in the cell line tested.

7) Finally, can it be distinguished from the data whether accumulated dsRNA in ADAR1 deficient cells directly activates basal OAS or perhaps a sensed by MDA5 to induce interferon causing a subsequent induction of OAS?

---

## [Author Response]

*We have now received comments on your manuscript from three experts in the specific area of ADAR1 biology. All agreed that the work was interesting, well done and in principle suitable for eLife. Nevertheless the following fairly minor points should be thoroughly addressed via revision.*

*1) Because ADAR1 technically belongs to the cytosine deaminase family of enzymes, a more appropriate title would be to substitute adenosine deaminase with the "double stranded RNA editing enzyme ADAR1 in a human cell line".*

Title has been changed as suggested.

*2) ADAR1 activity does not separate dsRNA into single stranded RNA but rather creates partially dsRNA. Please correct in the text and figure.*

We have modified Figure 1 to shown that ADAR editing produces partially dsRNA rather than ssRNA. We have also changed the wording in the legend to Figure 1. We have stated that ADAR1 “destabilizes dsRNA” in several places in the text.

*3) Please remove the word partially from "partially rescues" in the Abstract and also in the Introduction.*

This has been done as requested.

*4) Indicate the source and size of poly IC used.*

The sources of pIC are now indicated in Materials and methods. pIC (1.5-8 kb) was obtained from InvivoGen (see IncuCyte, cell death assay description), and used to obtain the data shown in Figure 3 and Figure 4. PIC (0.4 kb) was obtained from Millipore (see crystal violet death assay description) and used to obtain the data shown in Figure 3.

*5) Please quantify the Western blots in Figure 2 and Figure 3 and comment on the levels of PKR seen in interferon treated cells.*

We have quantified PKR bands in Figure 2 and Figure 5 as described in Materials and methods. (There are no Western blots in Figure 3). PKR expression, fold induced by IFN treatment over mock treated of various cell types is summarized in the figure legends and the data are shown in the [Supplementary-material SD1-data] and [Supplementary-material SD4-data] files.

PKR fails to be induced by IFN treatment in cells with ADAR1 or ADAR1 p150 KO. We do not currently understand why this occurs. We have added to the text; “However, it is noted that IFN treatment of ADAR1 deficient cells did not induce PKR expression, in contrast to ADAR1 expressing cells (Figure 2 and Figure 5). We do not currently know why this occurs.

*6) The reviewers agreed that the conclusions derived from this work should be tempered somewhat because results obtained primarily in a single cell line cannot necessary be extrapolated to the intact organism. Because mouse data indicates that the RIG-I/MDA5 is perhaps the primary pathway you could modify your conclusion to indicate that the OAS/RNase L pathway is an important contributor at least in the cell line tested.*

While Mda5 or Mavs deletion can rescue mouse embryonic lethality, the downstream pathway(s) that mediate lethality are not known (Figure 1). As described above, we have shown that while the OAS-RNase L pathway can be activated by endogenous dsRNA in the absence of MAVS in A549 cells, activation is more robust in the presence of MAVS (Figure 6 and Figure 7, explained above in item #7). Thus, RNase L may be the downstream pathway mediating lethality in mouse as well so that *Mavs* or *Mda5* KO leads to diminished RNase L activation and cell death and rescues embryonic lethality. Alternatively, there may be another effector downstream of MDA5/RIG-I-MAVS inducing cell death in mouse embryos. We have now stated in the Discussion; “It is possible that in the mouse embryo system RNase L, activated downstream of MDA5-MAVS also contributes to embryonic lethality. However, rescue of lethality due to *Adar1* KO by Mda5 or *Mavs* KO in the mouse embryo may be due to one or more other MAVS dependent activities.”

*7) Finally, can it be distinguished from the data whether accumulated dsRNA in ADAR1 deficient cells directly activates basal OAS or perhaps a sensed by MDA5 to induce interferon causing a subsequent induction of OAS?*

As depicted in Figure 1, OAS-RNase L can be either activated by dsRNA in a parallel pathway to the canonical MDA5-MAVS IFN induction pathway or as a MAVS dependent pathway. Our data show that RNase L can be activated in MAVS-ADAR1 DKO and MAVS-p150 DKO cells (Figure 7), indicating that MDA5 sensing of endogenous dsRNA is not required for activation of RNase L by endogenous dsRNA. However, MAVS expression (hence MDA5 sensing-MAVS signaling) does enhance RNase L activation presumably by up regulating OAS gene expression. This is supported by the data in Figure 7 induction) and also by the data in Figure 6 which show increased up regulation of IFN-λ and OAS2 following IFN treatment of RNase L-ADAR DKO compared to MAVS-ADAR1 DKO cells. In the original manuscript, we had discussed the effect of MAVS in enhancing 2-5A production and RNase L activation (tRNA degradation) through up regulation of OAS gene expression and have now added further text to clarify this point; “Thus 2-5A was produced and RNase L activated by endogenous RNA even in the absence of MAVS”.